# Effect of Electrical Conductivity of Nutrient Solution and Light Spectra on the Main Phytochemical Content of *Sonchus tenerrimus* L. Under Wild and Controlled Environments

**DOI:** 10.3390/plants14172811

**Published:** 2025-09-08

**Authors:** Tatiana P. L. Cunha-Chiamolera, Tarik Chileh-Chelh, Mohamed Ezzaitouni, José Luis Guil-Guerrero, Miguel Urrestarazu

**Affiliations:** 1Department of Agronomy, University of Almeria, La Cañada de San Urbano s/n, 04120 Almeria, Spain; tatiplcc@ual.es; 2Department of Agricultural Production, Faculty of Agronomic Sciences, University of Tarapaca, Arica 11315, Chile; 3Food Technology Division, ceiA3, University of Almeria, La Cañada de San Urbano s/n, 04120 Almeria, Spain; chileh@hotmail.es (T.C.-C.); mohamedezzaitouni6@gmail.com (M.E.); jlguil@ual.es (J.L.G.-G.)

**Keywords:** slender sow thistle, antioxidant activity, phenolics and flavonoids, stress tolerance, soilless culture, electrical conductivity, light spectrum

## Abstract

*Sonchus tenerrimus* L. is a wild leafy plant valued for its nutritional and functional properties. This study evaluated how different levels of electrical conductivity (EC) in nutrient solutions and lighting conditions affect the accumulation of bioactive compounds and growth performance in hydroponically cultivated *S. tenerrimus*. Plants were exposed to four EC treatments (1.2, 1.8, 2.4, and 3.0 dS m^−1^), four lighting regimens of natural light, and four artificial-lighting spectra. Total phenolic content (TPC), total flavonoid content (TFC), vitamin C, and antioxidant activity (via DPPH and ABTS assays) were measured. Principal Component Analysis (PCA) was used to assess the relationships among treatments and biochemical responses. The 2.4 dS m^−1^ EC level, particularly under natural light, led to the highest TPC, TFC, and antioxidant activity, indicating that moderate salinity enhances phytochemical production. Excessive EC (3.0 dS m^−1^) reduced antioxidant levels and plant growth, likely due to stress. Light conditions also influenced results, with natural light generally supporting greater bioactive accumulation and biomass than artificial lighting. These findings suggest that optimizing EC and light exposure can improve both the nutritional value and growth of *S. tenerrimus*. Future studies should explore the long-term effects, genotype-specific responses, and interaction of these factors with other environmental variables.

## 1. Introduction

Slender sow thistle (SST, *Sonchus tenerrimus* L.) is an annual plant species within the *Compositae* (also known as *Asteraceae*) family. A considerable number of species within this family are of commercial importance, as they are consumed worldwide due to their flavor and functional properties.

*Sonchus* species comprise annual, biennial, and perennial herbaceous plants [1]. The majority of these species are considered to be wild edible, with approximately 50 species distributed across Europe, Asia, and Africa [1]. However, only two species, *Sonchus asper* (L.) Hill and *Sonchus arvensis* L., are currently employed as natural drugs. The leaves of *Sonchus* spp. are used as a leafy vegetable because of their richness in protein, essential amino acids, vitamins, and minerals, thus enabling them to be used as healthy greens and for helping to ameliorate malnutrition-derived problems [2,3].

*Sonchus* species have a long history of utilization in traditional medicine, particularly in the form of infusions or decoctions. These preparations have been employed for the treatment of a wide range of ailments, including acute icterohepatitis, inflammation, cancer, diarrhea, rheumatism, and snake venom poisoning [4,5,6]. Recently, *Sonchus* species have attracted increased attention as a healthy dietary option, resulting in a growing body of scientific literature exploring their nutritional composition, chemical constituents, and biological activities.

There is a paucity of research reports on the phytochemicals and biological activities of SST. This wild green has been documented as containing elevated concentrations of protein, mineral nutrients, α-linolenic acid (ALA, 18:3*n*−3), vitamin C and carotenoids [7], luteolin and the luteolin-7-O-glucoside flavonoids, and chlorogenic and caffeic acids [8].

The sensory profile revealed a good general acceptance of *S. tenerrimus*, indicating that it could be included in the diet. In this species, cultivation increased the concentration of other compounds with important nutritional and health-promoting properties, such as sugars, organic acids, and β-carotene [9].

Many WEPs (wild edible plants) are being lost throughout generations in most Western cultures. The revalorization of wild vegetables as potential sources of functional food ingredients has the potential to increase their consumption in modern Western diets. A more profound understanding of wild edible plants, such as SST, has the potential to contribute to the enhancement of biodiversity and the sustainability of rural areas through the collection of these plants.

The biosynthesis and accumulation of phytochemicals in plants are profoundly influenced by soil composition and climatic conditions during growth. Nutrient uptake and the production of bioactive compounds in halophytes and other plants typically respond to both biotic and abiotic stresses, including salinity levels and light radiation [10]. Salinity has been identified as a significant abiotic stressor affecting crop plants [11]. While there has been extensive research on the effects of salinity in nutrient solutions on nutrient composition and phytochemical accumulation in various crops [12], there is limited information regarding the effects of salinity on the phytochemical properties of wild edible plants, such as *S. tenerrimus* L.

Light conditions exert a substantial influence on the morphology and physiology of plants, as well as on the biosynthesis and accumulation of phytochemicals, particularly within controlled growth environments [13]. Furthermore, the light intensity of LEDs can be precisely adjusted to optimize both plant growth and phytochemical composition [14]. Consequently, the utilization of LEDs has been on the rise in diverse horticultural domains, including studies on photoperiod lighting for greenhouses [15]. LED lighting systems offer significant advantages over traditional lighting due to their spectral composition, durability, wavelength specificity, low radiant heat, and energy efficiency [10]. In this context, the influence of different LED lights on wild plants has been evaluated, such as in Ice Plant (*Mesembryanthemum crystallinum* L.), Slender Amaranth (*Amaranthus viridis* L.), and London Rocket (*Sisymbrium irio* L.). The results indicate that specific LED lamp spectra have a positive effect on both vegetative growth and phytochemical composition, particularly when compared to white LED lamp [11,12,16].

The antiproliferative potential of *Sonchus* species has received scant attention from researchers, although preliminary studies have demonstrated encouraging cytotoxic effects that are attributed to the high content of phenolic compounds, flavonoids, and other bioactive metabolites in these species. In a recent study, the essential oil of *Sonchus oleraceus* exhibited significant cytotoxic activity against HepG2 cells, with an IC_50_ value of 136.02 µg mL^−1^ in the MTT assay [17]. Furthermore, the study investigated the cytotoxic effects of leaf extracts in n-hexane, dichloromethane, ethyl acetate, and n-butanol on three cancer cell lines: MCF-7 (breast), HCT-116 (colon), and HepG2 (liver). The results indicated that the n-hexane fraction exhibited potent cytotoxic activity against all tested cell lines, while the dichloromethane fraction showed particularly strong activity against HCT-116 cells (IC_50_ 5.42 μg mL^−1^) [18].

Despite the global consumption of *Sonchus tenerrimus* (SST) leaves, there is a paucity of research concerning its adaptability to diverse cultivation conditions. A more comprehensive understanding of their cultivation potential is therefore essential if SST is to be considered as a regular food source. Moreover, there is a gap in the scientific data concerning the phytochemicals and biological activities of SST, emphasizing the necessity for further research in this area. In this context, the present study focused on unraveling the phytochemical composition, antioxidant activity, and in vitro antitumor activity against colorectal cancer cells of SST leaves, both wild-harvested and cultivated under controlled conditions. The primary objective of this study was to explore the potential health benefits of these leaves, evaluating their capacity to contribute to human health and their potential application as a source of bioactive compounds.

## 2. Results

### 2.1. Effect of Salinity and Light on Growth Parameters in Cultivated S. tenerrimus

#### 2.1.1. Fertigation Parameters

The volume fraction of fertigation to drainage was maintained between 0.2 and 0.3 to ensure that the matric potential of water in the substrate remained constant [19]. In the electrical conductivity (EC) experiment, a significant variation in drainage pH was observed among the different treatments over the five-week cultivation period. The maximum recorded value was pH 7.5, representing an increase of up to 1.7 units above the initial fertigation solution pH of 5.8 (Appendix A). Despite this variability, no statistically significant differences were found in the average pH values among treatments with different EC levels (Appendix A). It is worth noting that under vegetative growth conditions of *S. tenerrimus*, drainage pH levels are expected to increase by up to two units above the fertigation pH value, as reported by Urrestarazu and Carrasco [20]. In this context, the average pH values recorded in the present study remain within the acceptable range for optimal crop development.

The EC of the drainage recorded during the crop exhibited a general trend of progressive increase, reaching values that exceeded the fertigation solution level by more than two units. This trend suggests a gradual rise in nutrient concentration in the growing medium, likely due to the accumulation of salts or nutrients not fully absorbed by the plants (Appendix A). The data show that mean EC values in the drainage varied significantly, directly influenced by the different applied EC levels. The highest drainage values were observed in treatments with EC of 2.4 to 3.0 dS m^−1^ (Appendix A), highlighting a direct correlation between the salinity of the nutrient solution and salt accumulation in the substrate. Despite these differences, all mean drainage EC values remained within agronomically acceptable and desirable limits for the crop. In general, drainage EC values stayed between 1 and 2 dS m^−1^ above the initial fertigation level, suggesting that the crop exhibits a remarkable tolerance to moderate fluctuations in root-zone salinity without manifesting adverse physiological effects. These results are in agreement with those reported by Urrestarazu and Carrasco [20], who noted that this crop can perform adequately within a relatively broad EC range without compromising yield or product quality.

In relation to the illumination trial, no significant differences in the parameters associated with fertigation monitoring were observed throughout the growing season. The pH values remained relatively stable during the first three weeks, followed by a slight decrease in the last two weeks (Appendix A). Despite this variation, mean pH values did not show statistically significant differences between treatments (Appendix A). In general, pH levels recorded during cultivation remained within acceptable ranges, being around one or two units above the reference value of the fertigation solution (pH 5.8), which is consistent with normal growing conditions in soilless systems [20,21].

EC values remained relatively stable throughout the cultivation period, with a slight increase observed under lighting treatments L1 and L2 during the final week (Appendix A). This increase may be associated with a higher salt concentration in the drainage, potentially due to reduced water uptake by the plants in the later stages of growth or light spectrum-induced changes in transpiration. Despite these occasional variations, the mean EC values in the drainage did not show statistically significant differences among the treatments evaluated (Appendix A), indicating that the type of illumination had no substantial effect on the salinity dynamics of the system. These findings suggest that the ionic balance of the growing medium remained stable, regardless of the light source applied.

#### 2.1.2. Fertigation Uptake and Growth Parameters

There was a statistically significant difference in fertigation uptake among the treatments in the EC experiment (Figure 1A), with the lowest uptake observed at an EC level of 3.0 dS m^−1^. No significant differences were found among treatments with EC levels ranging from 1.8 to 2.4 dS m^−1^, indicating that moderate salinity did not substantially affect water and nutrient absorption. However, the increase in fertigation salinity to 3.0 dS m^−1^ led to a 17.5% reduction in uptake, suggesting that excessive salt concentration can impair the plant’s ability to absorb water effectively. This pattern of fertigation volume uptake is consistent with the classical growth response model described by Sonneveld and Voogt [22], in which high salinity levels negatively impact plant–water relations. Similar results were also reported for Ice Plant [11], further supporting the notion that elevated electrical conductivity can limit nutrient solution absorption, particularly beyond the crop’s salinity tolerance threshold.

However, an increase in nitrate and potassium uptake was observed as the concentration of these ions increased in the nutrient solution, even with the progressive rise in EC (Figure 1B,C). Although a significant decrease in potassium uptake was detected at the highest EC level of 3.0 dS m^−1^. This behavior suggests that, within certain thresholds, plants are able to actively absorb these nutrients even under conditions of elevated salinity. These findings are consistent with those reported by Gallegos-Cedillo et al. [23], who also observed a continuous increase in nitrate and potassium uptake as their concentrations in the nutrient solution increased, regardless of the EC level. This pattern supports the hypothesis that the uptake of essential ions such as NO_3_^−^ and K^+^ is primarily driven by their availability in the rhizosphere, rather than by moderate variations in total salinity—at least up to a physiological tolerance threshold specific to the crop.

Nitrate and potassium uptake in *S. tenerrimus* followed a pattern similar to water absorption, with the lowest values recorded under treatment L4 (Figure 2). In general, treatments using horticulture-specific light fixtures resulted in higher water and nutrient uptake compared to the control treatment with standard LED lighting (L4). An exception to this trend was observed in treatment L1, which showed reduced potassium uptake relative to the other horticultural lighting treatments. Nitrate uptake closely mirrored the pattern observed in water absorption, suggesting a functional relationship between water and nutrient transport in response to light spectral quality. Nevertheless, treatment L4 consistently exhibited the lowest uptake values across all three parameters, with reductions of 33.4% in water uptake, 27.2% in nitrate uptake, and 66.6% in potassium uptake compared to the horticultural lighting treatments. These results contrast with the findings of Peçanha et al. [19], who reported no significant differences in water and nutrient uptake when using the same types of lamps for the rooting of lavender plants. This discrepancy suggests that plant responses to light quality may be species-specific or influenced by the developmental stage of the crop.

Growth response, measured in terms of fresh and dry weight, aligned with the classical salinity response model proposed by Mass and Hoffman [24] and the production model described by Sonneveld and Voogt [22]. According to these models, crops exhibit a tolerance phase up to a certain EC threshold, beyond which yield declines progressively. In this study, the threshold was identified at an EC level of 2.4 dS m^−1^, beyond which a significant reduction in yield was observed (Figure 3). Specifically, yield decreased by 28.2% for each additional unit increase in salinity beyond the threshold, indicating that high salinity levels exert osmotic and/or ionic stress that negatively affects biomass accumulation. In contrast, no statistically significant differences in yield were found among treatments with EC levels ranging from 1.8 to 3.0 dS m^−1^, suggesting that *S. tenerrimus* exhibits moderate tolerance to salinity within this range.

Higher biomass accumulation in *S. tenerrimus* was directly associated with higher water uptake (Figure 4). Treatments using lamps specifically designed for horticultural applications promoted significantly higher growth compared to the control treatment, which employed standard LED lighting (L4). Both fresh weight and dry weight yields were notably higher in these specialized treatments, indicating that the spectral quality of light directly influences crop physiological efficiency. This improvement in biomass suggests a higher photosynthetic rate and better functioning of water and nutrient uptake mechanisms under horticulturally optimized lighting conditions.

### 2.2. Phytochemical Characterization

The results of moisture content, total carotenoids, vitamin C, total phenolics, flavonoids, and antioxidant activity evaluated via DPPH and ABTS methods are presented in Table 1.

#### 2.2.1. Moisture

The moisture content exhibited variability across the wild samples, ranging from 85.6 g 100 g^−1^ in WA to 87.0 g 100 g^−1^ in WU. Conversely, cultivated plants subjected to varying salinity conditions exhibited mean moisture values ranging from 82.4 to 85.3 g 100 g^−1^ in C3 and C2, respectively. Furthermore, the moisture content of plants exposed to different light treatments demonstrated fluctuations, with values ranging from 83.3 (L3) to 86.6 g 100 g^−1^ under L2 lamps (see Table 1 for further details).

#### 2.2.2. Total Carotenoids

The carotenoid concentrations are presented in Table 1. The lowest concentrations were observed in cultivated plants subjected to different light treatments, ranging from 32.0 to 44.7 mg 100 g^−1^ FW in L1 and L4 plants. Conversely, an increase in salinity resulted in higher total carotenoid levels, ranging from 35.9 to 48.2 mg 100 g^−1^ FW in C1 and C4 plants. Furthermore, leaves from wild plants exhibited significantly greater concentrations compared to those of cultivated plants, ranging from 45.3 to 61.5 mg 100 g^−1^ FW in WD and WA plants.

#### 2.2.3. Vitamin C

As illustrated in Table 1, a comprehensive dataset pertaining to the vitamin C levels in diverse samples of *S. tenerrimus* is presented, encompassing both wild and cultivated states that have been subjected to various saline and light stress treatments. The data demonstrate a wide range of vitamin C concentrations, from a minimum of 53.2 mg 100 g^−1^ FW in WD to a maximum of 105.1 mg 100 g^−1^ FW in C4. A salient finding was the marked differences observed between the vitamin C levels in wild samples and those cultivated under saline and light stress conditions. This analysis is of paramount importance in elucidating the diverse nutritional profile of this plant and its potential implications for health and nutrition.

#### 2.2.4. Total Phenols and Flavonoids

As illustrated in Table 1, a comprehensive overview is provided of the total phenolic compound (TPC) and total flavonoid (TFC) contents in various samples of *S. tenerrimus*, spanning both wild and cultivated sources, and subjected to diverse salt and light stress treatments. The TPC values ranged from 197.0 mg GAE 100 g^−1^ FW in L4 to 430.6 mg GAE 100 g^−1^ FW in WD (Table 1). Conversely, TFC exhibited notable variation, with the WA (wild edible plants) sample demonstrating the highest value at 286.5 mg QE 100 g^−1^ FW, while the L4 sample exhibited the lowest value at 72.1 mg QE 100 g^−1^ FW (Table 1). These bioactive compounds are of significant interest due to their potential health benefits and their contribution to the antioxidant and nutritional properties of the plant.

#### 2.2.5. Antioxidant Activity

The antioxidant activity of various samples of *S. tenerrimus* is detailed in Table 1, encompassing both wild and cultivated states, and subjected to diverse salt and light stress treatments. The range of DPPH values was from 1.2 to 3.0 mmol TE 100 g^−1^ DW, with the highest recorded in sample L1 and the lowest in L3. Conversely, the antioxidant activity measured by the ABTS-exhibited variations between 0.6 and 3.0 mmol TE 100 g^−1^ DW, with the highest recorded in C3 and the lowest in L4 (Table 1). This analysis is pivotal for comprehending the variability in the antioxidant capacity of this plant under varying environmental conditions, alongside its potential applications in the pharmaceutical and food industries.

### 2.3. Principal Component Analysis (PCA)

This statistic was performed to explore potential relationships among the analyzed parameters and sample characteristics. The analysis included all variables represented in the corresponding plot. A total of 14 components were extracted, collectively accounting for 100.0% of the total variance in the dataset. The first two principal components (PC1 and PC2) explained 43.0% and 30.3% of the total variance, respectively, together representing 70.3% of the overall variability. Among the various graphical outputs provided by the PCA, the biplot offered the most informative visualization. In the biplot, the horizontal axis corresponds to PC1 and the vertical axis to PC2. The geometric distribution of the data points suggests that the samples can be grouped based on similarities in the measured variables. Figure 5 presents the projection of the PCA biplot onto the plane defined by PC1 and PC2.

## 3. Discussion

### 3.1. Effect of Salinity and Light on Growth Parameters in Cultivated S. tenerrimus

#### 3.1.1. Fertigation Parameters

The drainage volume fraction, electrical conductivity (EC), and pH are commonly used parameters for monitoring and managing the nutrient solution and scheduling fertigation [19]. The influence of pH on plant growth and nutrient uptake in horticultural crops has been widely recognized for decades [25,26] and remains relevant in current research [10].

In the EC experiment, the pH of the drainage water varied between 0.5 and 1.7 units above the initial value of the nutrient solution (5.8; Appendix A). According to Ferrón-Carrillo et al. [27], under vegetative growth conditions in species such as *S. tenerrimus*, pH fluctuations of up to two units above the reference value can be expected. This suggests that the observed changes remained within physiologically acceptable limits for optimal plant development. This indicates that, as a leafy vegetable, the rate of nitrate uptake is higher than that of potassium, thereby lowering the pH value.

The mean EC values of the drainage in Experiment 1 also remained within acceptable ranges, with values from 1 to 2 dS m^−1^ higher than those of the applied nutrient solution (Appendix A). These standard conditions reflect increased metabolic activity at the root level, particularly in the uptake of key nutrients such as potassium and nitrate, in proportion to the volume of water absorbed by the plants [28].

In the lighting experiment, no significant differences were observed in drainage pH or EC values among treatments. This result contrasts with findings reported by Ferrón-Carrillo et al. [10], who documented increased growth and yield in pepper and tomato plants grown under agricultural LED lighting compared to control treatments with standard LED fixtures.

#### 3.1.2. Fertigation Uptake and Growth Parameters

In relation to the EC experiments, the results indicate that treatments with EC levels higher than 2.4 dS m^−1^ caused a significant reduction in both water uptake and growth of *S. tenerrimus*, measured through the fresh and dry weight of biomass. This decrease in development suggests a high sensitivity of the species to high salinity conditions, possibly due to osmotic and ionic effects that interfere with the physiological processes of cellular absorption and expansion. These findings are consistent with previous studies in *Sisymbrium irio*, where a reduction in water uptake and biomass accumulation has also been observed under saline conditions [16].

Water uptake, a parameter highly correlated with plant growth and yield [21,27], showed significant reductions under higher salinity levels in the nutrient solution. Under similar growing conditions, *Mesembryanthemum crystallinum* has exhibited a markedly different response: Rincón-Cervera et al. [11] reported optimal EC levels well above 1.8 dS m^−1^ without compromising water uptake or productivity. While this species is known for its high salt tolerance, *S. tenerrimus* behaved as a salt-sensitive species, showing a clear reduction in both water absorption and biomass accumulation under elevated salinity conditions.

Regarding mineral nutrition, nitrate and potassium are the main anions and cations absorbed by higher plants, making them reliable indicators of root efficiency in nutrient uptake [10,25,29]. The patterns observed in nitrate and potassium uptake across the different EC treatments reinforce their central role in monitoring plant growth and fertigation efficiency.

Studies on tomato plants by Gallegos-Cedillo et al. [23] found that nitrate and potassium absorption increased with higher concentrations of these nutrients in solutions with EC levels ranging from 2.4 to 3.0 dS m^−1^. In general, greater nutrient uptake is directly associated with increased availability in the fertigation solution, facilitating both active and passive ion transport into the plant [20,23]. However, this relationship is also modulated by the species-specific tolerance to salinity, as evidenced by the sensitivity of *S. tenerrimus* observed in this study.

The spectral composition of light not only plays a fundamental role in biomass growth but also significantly influences stomatal functionality and the overall water-use efficiency of plants [11]. These factors are crucial for the physiological development and agronomic performance of crops. In the lighting experiment, the treatment corresponding to L4 (L18 T8 Roblan^®^) showed the lowest values in terms of water and nutrient uptake, suggesting reduced physiological efficiency under this specific light spectrum. This finding underscores the importance of selecting appropriate lighting conditions in controlled environments. Spalholz et al. [30] emphasize that both the spectral composition and light intensity are key variables for maximizing crop performance. According to these authors, the precise adjustment of artificial light characteristics can significantly enhance various aspects of plant growth, such as photosynthesis, water-use efficiency, nutrient absorption, and, ultimately, overall productivity.

Moreover, lamps specifically designed for horticultural use offer a broader and more balanced light spectrum, with an increased proportion of wavelengths in the blue light range. This more closely simulates natural sunlight and enhances energy absorption by photosynthetic pigments, thereby promoting more efficient plant growth compared to the results observed under the L4 treatment. These findings further highlight the critical role that both spectral composition and light intensity play in horticultural systems. As emphasized by Spalholz et al. [30], light conditions tailored to the physiological needs of plants can optimize various aspects of growth and overall productivity. Similarly, Nájera and Urrestarazu [31] reported significant improvements in plant development when using agricultural-grade LED lighting, especially the case when the proportion of far red in the spectrum is increased in a light fixture. These systems are specifically engineered with intensity and spectral characteristics designed to maximize photosynthetic and metabolic processes in plants grown in controlled environments.

### 3.2. Total Carotenoids

Carotenoids are defined as essential organic pigments in plant metabolism, where they play a key role as free radical scavengers and protectors against oxidative stress [32]. In addition to their role as antioxidants, certain carotenoids, such as α-carotene, β-carotene, and β-cryptoxanthin, have been found to exhibit provitamin A activity. These carotenoids are vital for sustaining visual function and other physiological processes that are contingent on vitamin A.

The carotenoid content of wild *S. tenerrimus* samples was found to be considerably higher than that of cultivated samples. These liposoluble pigments are imperative for photosynthesis, functioning not only as light-harvesting molecules but also as crucial antioxidants that safeguard plant tissues from oxidative damage induced by environmental stressors. The greater accumulation observed in wild plants may be attributed to their exposure to more demanding natural conditions, such as high solar irradiance, temperature fluctuations, and water scarcity. These conditions promote carotenoid biosynthesis as a photoprotective response [33,34].

Among the cultivated plants, the saline treatment C4 (3.0 dS m^−1^) exhibited the highest carotenoid content, suggesting that moderate-to-high salinity levels can trigger adaptive antioxidant responses, as previously reported in other leafy species by López-Berenguer et al. [35]. Regarding the application of light treatments, L4 resulted in a heightened accumulation of carotenoids in comparison to alternative artificial light conditions. This phenomenon can be attributed to the distinct spectral composition of the L4, which has been observed to stimulate the expression of biosynthetic pathways involved in the production of secondary pigments [36].

In this study, the elevated levels of carotenoids observed in *S. tenerrimus* under conditions of elevated salinity suggest that the carotenoid biosynthetic pathway was activated as an adaptive response to stress. It has been well documented that exposure to high salinity levels results in an increase in the generation of ROS within plant tissues. This increase in ROS generation has been shown to stimulate carotenoid biosynthesis as part of the plant’s antioxidant defense system [37]. Conversely, the levels of carotenoids in *S. tenerrimus* cultivated under different artificial light sources exhibited a comparatively narrow range of variation (31.6–44.7 mg 100 g^−1^ FW), a finding that corroborates previous studies on indoor-grown plants under red and blue LED light combinations [38,39].

### 3.3. Vitamin C

As a water-soluble micronutrient, vitamin C is essential for the human body and is known for its potent antioxidant activity and its role as a cofactor in various enzymatic reactions. It is involved in vital biological functions, including collagen synthesis, iron absorption, and immune system strengthening, in addition to protecting cells against oxidative damage caused by free radicals [40]. As the human body is incapable of synthesizing it, it must be obtained through the diet. The recommended daily intake of vitamin C for adults is 75 to 90 mg; however, this amount may vary depending on age, sex, health status, and other individual factors [40].

The analysis revealed significant variability among the treatments, with the highest content recorded in the saline C4 treatment (105.1 mg 100 g^−1^ FW), which significantly exceeded the levels observed in wild samples. This finding is of significance, as ascorbic acid has multiple functions in plant metabolism, including the neutralization of ROS, the regeneration of other antioxidants and the modulation of enzymes of the antioxidant cycle [41,42]. Previous studies have reported that cultivated plants often exhibit higher or comparable concentrations of vitamin C than their wild counterparts. Examples include *Mesembryanthemum nodiflorum* L., *Suaeda maritima* (L.) Dumort., *Sarcocornia fruticosa* L., *Amaranthus viridis* L., and *Sisymbrium irio* L. This pattern may be explained by the absence of metabolic stress associated with nutrient or water deficiency, which frequently occurs under natural growing conditions and leads to highly variable vitamin C concentrations [43].

The wild samples, although not reaching the levels of C4, also exhibited significant amounts, as observed in the WT (89.4 mg 100 g^−1^ FW), suggesting that natural environmental factors also promote the accumulation of this vitamin. It is important to note that vitamin C, due to its water-soluble nature and susceptibility to oxidation, can be influenced by multiple factors during plant development, including irradiation, temperature, and water availability [44,45].

### 3.4. Total Phenols and Flavonoids

TPCs are widely recognized for their strong antioxidant activity, and they reached significantly high concentrations in both wild and cultivated *S. tenerrimus* plants. The wild sample WD exhibited the highest value (430.6 mg GAE 100 g^−1^ FW), suggesting an intense metabolic response likely associated with a natural environment characterized by greater oxidative stress. The elevated accumulation of phenolics in wild plants is consistent with numerous studies reporting enhanced biosynthesis of these compounds under adverse environmental conditions [46]. In a similar manner, TFC, a subclass of phenolic compounds, also attained their maximum levels in wild samples, with WD once again being a notable outlier at 286.5 mg QE 100 g^−1^ FW. This finding is consistent with the existing literature, which attributes a protective role to flavonoids against stress factors such as ultraviolet radiation, dehydration, and pathogen attack [47,48].

In cultivated plants, the saline treatment C4 also promoted a high level of TPC, reaching 380.8 mg GAE 100 g^−1^ FW. This suggests that controlled salt stress can mimic the effects of natural environmental stress and stimulate phenolic biosynthesis. Regarding TFC, plants cultivated under saline conditions exhibited intermediate levels, with notable values observed in treatments C1 and C2 (~166–170 mg QE 100 g^−1^ FW). This trend may indicate a partial inhibition of flavonoid biosynthetic pathways as salinity increases, or a possible reallocation of secondary metabolism toward other classes of bioactive compounds [49].

The impact of light treatments on the accumulation of TPC was found to be generally low, with the highest value observed in L2 (305.6 mg GAE 100 g^−1^ FW) and the lowest in L4 (197.0 mg GAE 100 g^−1^ FW). The results of this study indicate that the spectral quality of light may exert a partial modulatory effect on the biosynthetic pathways involved in phenolic production, as reported by Li and Kubota [50]. Regarding TFC, the efficacy of artificial light was found to be even more limited, a phenomenon that may be ascribed to the absence of ultraviolet (UV) radiation in the light sources employed. UV light, a fundamental element in natural conditions, has been demonstrated to play a pivotal role in the activation of specific genes involved in flavonoid biosynthesis [51,52].

### 3.5. Antioxidant Activity

As posited by Calvo et al. [53], phenolic compounds have been demonstrated to play a pivotal role in protecting plants against oxidative stress induced by ROS, with a generally observed positive correlation between phenolic content and the antioxidant activity of plant extracts. In this study, the ABTS and DPPH assays were utilized to evaluate antioxidant activity based on the extracts’ capacity to scavenge free radicals, given the high sensitivity of both methods [54]. The ABTS assay, which involves the generation of a specific free radical, is particularly useful for quantifying antioxidant activity in both hydrophilic and lipophilic systems and is therefore considered more effective in detecting antioxidant capacity across a wide range of plant tissues. Conversely, the DPPH assay demonstrates higher efficacy in hydrophobic environments [55].

The antioxidant capacity, as determined by the DPPH and ABTS methods, exhibited significant variability among the treatments, with no discernible direct or linear correlation with the levels of individual compounds. In the DPPH assay, treatment L1 (NS1 light) exhibited the highest value (3.0 mmol TE 100 g^−1^ DW), whereas in the ABTS assay, treatments C2 (1.8 dS m^−1^) and C3 (2.4 dS m^−1^) stood out, with values of 2.9 and 3.0 mmol TE 100 g^−1^ DW, respectively. These findings suggest that different antioxidant compounds exhibit specific affinities for the radicals assessed in each assay, as previously described by Brand-Williams et al. [56].

The relatively elevated values observed in saline crops and under specific light treatments suggest that controlled conditions can also induce an effective antioxidant response; however, this response is likely to be of a different nature compared to that of wild plants, where a more complex response was detected, involving a combination of carotenoids, flavonoids, and phenolic compounds. This diversity in antioxidant profiles may confer greater in vivo biological efficacy, which is particularly relevant from a nutraceutical and functional perspective.

It is important to mention the significant influence of the extraction solvent on the antioxidant capacity of the extracts [57]. This factor, in conjunction with the influence of biotic and abiotic conditions during plant growth, may provide a rationale for the discrepancies in antioxidant activity of *S. tenerrimus* extracts reported in various studies.

### 3.6. Principal Component Analysis

In Figure 5, this procedure was applied to the cultivated samples, allowing the monitoring of parameters that influence plant development.

In Figure 2, it can be noted that all samples from the EC experiment (C1–C4) have positive scores for PC1; and all samples from the lighting trial have negative scores for PC1. In this plot, it can be noted how different variables influencing the measured parameters, namely DW, ABTS, TFC, TPC, water, and K, have positive loads for PC1 and PC2. TPC, TCC, and ABTS form a group in the plot, thus indicating that the antioxidant activity measured by ABTS was mainly exercised by TPC and TFC. As previously stated, the ABTS is useful for quantifying antioxidant activity in both hydrophilic and lipophilic systems and is therefore considered more effective in detecting antioxidant capacity. Note how the antioxidant activity measured by DPPH is in the opposite quadrant, that is, presenting negative loads for both components, PC1 and PC2. Only carotenoids are close to DPPH, which can be explained considering that, due to their lipophilic nature, they probably exerted antioxidant activity capable of being measured by this methodology.

EC, nitrate, vitamin C, and carotenoids have positive scores for PC1 and negative for PC2. EC, which is an applied variable, thus had influence on nitrate accumulation, as well as vitamin C and carotenoids biosynthesis.

In this regard, it has been reported that high EC in nutrient solutions can enhance the biosynthesis of vitamin C and carotenoids, though optimal levels vary by crop. For example, bell peppers and tomatoes exhibited increased vitamin C content when grown under EC levels of 2.0 dS m^−1^ or higher [58,59]. Similarly, kohlrabi showed optimal vitamin C accumulation at EC levels between 2.3 and 2.9 dS m^−1^ [60].

Carotenoid content is also positively influenced by EC. For instance, in carrots, pulsed electric field (PEF) treatments—related to electric conductivity—enhanced levels of phytoene and β-carotene [61]. Likewise, increased EC promoted higher β-carotene and lycopene concentrations in bell peppers and tomatoes [58,59].

However, high EC can lead to greater nitrate accumulation in plant tissues. Lettuce grown at 10 dS m^−1^ showed elevated nitrate content [62]. Conversely, reducing EC or applying beneficial chloride nutrition improved nitrate assimilation and lowered leaf nitrate levels in butterhead lettuce [63,64].

Finally, the samples from the lighting experiment (L1–L4) show negative scores for PC2. This means that the application of any of the checked lamps lacks the tendency to induce any biocompound synthesis or bioactivity development.

## 4. Materials and Methods

### 4.1. Solvents and Reagents

It is to be noted that, unless stated otherwise, all reagents and solvents utilized in the present study were obtained from Merck (Madrid, Spain; see Appendix A for details).

### 4.2. Samples and Growth Conditions

Information regarding *S. tenerrimus* specimens studied in this work is shown in Table 2. Leaves from wild plants were collected from the locations listed in Table 2, whereas cultivated plants were grown in a soilless system with LED lamps at the University of Almeria. Cultivation took place in controlled growth chambers (10 × 2.5 m) between March and May 2022. Environmental conditions were maintained at a 16/8 h light/dark photoperiod, temperatures of 20–22 °C, relative humidity of 80–85%, and a photosynthetic photon flux density of 250 μmol m^−2^ s^−1^ (400–700 nm). Detailed information on growth, fertigation, and lighting conditions is available in Appendix A. Two independent experiments were conducted. In the first, three treatments with different nutrient solution electrical conductivities (ECs) were applied: 1.2 (C1), 1.8 (C2), 2.4 (C3), and 3.0 (C4) dS m^−1^. Illumination was provided by a commercial Roblan^®^ L18 T8 white LED lamp (Toledo, Spain). The composition of the nutrient solutions is presented in Appendix A. In the second experiment, four different LED lamps (treatments L1 to L4; see Table 2) were evaluated. The spectral profiles of each lamp are shown in Appendix A. A constant EC of 2.0 dS m^−1^ was used for all treatments. In both experiments, the pH of the nutrient solutions was adjusted to 5.8 using nitric acid.

*S. tenerrimus* plants were harvested 30 days after transplanting, fresh weight (FW) was measured, and then DW was obtained by placing the material in an oven (Thermo Scientific Heratherm, Waltham, MA, USA) at 85 °C until a constant weight was reached to determine the moisture content. A precision analytical balance (model AX 124/E, OHAUS Corporation, Parsippany, NJ, USA) was used, expressing the result as g plant ^−1^. Once in the laboratory, samples were labeled, weighed (2 g), measured, and placed in a glass desiccator until analysis.

### 4.3. Total Carotenoids

Total carotenoid content was determined as described by Whyte [65]; the process involves the saponification of the biomass with KOH, followed by the extraction of carotenoids with diethyl ether and subsequent resuspension in acetone. Then, ~200 mg of fresh leaf material was weighed and treated with 1 mL of 60% (*w*/*w*) KOH, followed by sonication for 10 min. The homogenate was then heated to 40 °C for 40 min, vigorously stirred, and left to stand in darkness at 4 °C for 24 h. It was then subjected to ultrasonication for 20 min, and the pigments were extracted with 1 mL of diethyl ether after centrifugation at 2000× *g* for 5 min. A second extraction with diethyl ether was performed, followed by drying under nitrogen flow, and finally resuspended in 5 mL of acetone. Optical density was measured at a wavelength of 444 nm in a 1 cm glass cuvette using acetone as a blank. The quantification of total carotenoids was achieved using a calibration curve, with a β-carotene standard in acetone (0–20 ppm) serving as the standard. The results obtained are expressed in mg of carotenoids per 100 g of fresh weight (FW).

### 4.4. Extraction and Quantification of Vitamin C

The determination of vitamin C (L-ascorbic acid) content was conducted in accordance with Volden et al. [66], with minor adaptations. In summary, ~0.5 g of fresh *S. tenerrimus* sample was extracted with 5 mL of 1% (*w*/*v*) oxalic acid. The mixture was then subjected to centrifugation at 2000× *g* for 10 min. Later on, the sample was filtered through filter paper, the resulting filtrate was collected, and 1 *mL* of the vitamin C extract was filtered again through a 0.22 µm Millipore before chromatographic analysis.

The analysis of ascorbic acid was conducted using a Finnigan Surveyor chromatograph, which was equipped with a diode-array detector (DAD) and a reverse-phase C18 column (Luna^®^ Omega C18, 250 mm × 4.6 mm i.d., 3 µm particle size) manufactured by Phenomenex (Orlando, FL, USA). The mobile phase consisted of 5% (*v*/*v*) methanol/water (0.1% oxalic acid in water) in isocratic mode. The wavelength detection was set at 254 nm. The flow rate of the mobile phase was 0.4 mL min^−1^, and the volume of the injected sample was 10 µL. The total duration of the analysis was 15 min. Ascorbic acid was quantified by external calibration, with results expressed in milligrams per 100 g FW. All data are presented as mean ± standard deviation of triplicate analysis.

### 4.5. Extraction of Phenolic Compounds

The extraction and analysis of phenolic compounds from *S. tenerrimus* samples was conducted in accordance with the methodology outlined by Lyashenko et al. [67], with certain modifications. Approximately 1 g of sample was weighed; 10 mL of a solution composed of ethanol/water (96:4, *v*/*v*) was added; the sample was then macerated and subjected to centrifugation at 4000× *g* rpm for 10 min; and, subsequently, the sample was filtered through filter paper, and the resulting filtrate was collected. Then, 1 mL of the phenolic extract was taken and filtered again through a 0.22 μm membrane filter before chromatographic analysis.

### 4.6. Determination of the Total Phenolic Content

The total phenolic content (TPC) was measured using the Folin–Ciocâlteu assay, as developed by Singleton et al. [68], with minor modifications. In summary, 10 μL of *S. tenerrimus* phenolic extracts (the preparation of which is fully detailed in Appendix A, 0.79 mL of MiliQ water, and 50 μL of Folin–Ciocâlteu reagent were mixed, vortexed, and left to stand for 5 min at room temperature. Subsequently, 150 μL of a 20% sodium carbonate solution was added and vortexed. Following this, the mixture was subjected to incubation at room temperature for a period of 2 h in conditions of darkness. Thereafter, the optical density of the mixture was measured at a wavelength of 765 nm using a UV-VIS spectrophotometer (Helios, Thermo Spectronic, Cambridge, UK). The blank, which had been prepared as described above but without standard solutions, served as the control. The results were expressed as mg of gallic acid equivalents (GAEs) per 100 g of sample using a standard curve of GAE (ranging from 50 to 900 μg mL^−^^1^). It is important to note that the determinations were conducted in triplicate.

### 4.7. Determination of Total Flavonoid Content

The total flavonoid content (TFC) of the phenolic extract of *S. tenerrimus* samples was determined according to the method outlined by Zou et al. [69], with certain modifications. Briefly, 0.5 mL of the phenolic extracts were taken, and 150 μL of 5% NaNO_2_ solution was added. Following a period of 5 min, 150 μL of a 10% AlCl_3_ solution was added to the mixture, which was then kept at room temperature for a further 5 min. This was followed by the addition of 0.7 mL of 1 M NaOH. The resulting solution was thoroughly mixed, and the spectrophotometric analysis was conducted at 510 nm using a UV-VIS spectrophotometer (Helios, Thermo Spectronic, Cambridge, UK) against a blank, which was prepared as described above but without standard solutions. The results were expressed as mg of quercetin equivalents (QEs) per 100 g of sample using a standard curve of quercetin (ranging from 10 to 500 mg per mL). Determinations were conducted in triplicate.

### 4.8. Determination of the Antioxidant Activity

Sample extraction was performed according to the established protocol for the analysis of total phenolic content (TPC) and total flavonoid content (TFC). The extraction solvent consisted of ethanol and water in a 96:4 (*v*/*v*) ratio.

The antioxidant activity was evaluated using both the ABTS and DPPH assays.

For the ABTS assay, the ABTS^•+^ radical cation (2,2′-azinobis(3-ethylbenzothiazoline-6-sulfonic acid)) was generated by reacting a 7 mM ABTS solution in methanol with a 2.45 mM potassium persulfate solution prepared in phosphate buffer (pH 7.0) at a 1:1 (*v*/*v*) ratio. The mixture was incubated at 25 °C in the dark for 16 h. After incubation, 1950 μL of the resulting ABTS^•+^ solution was mixed with 50 μL of the ethanolic extract. The mixture was vortexed at 2000 rpm for 1 min and incubated in the dark at 25 °C for 7 min. Absorbance was then measured at 734 nm [70].

The DPPH assay was performed with slight modifications to the method described by Skenderidis et al. [71]. A 0.25 mM stock solution of DPPH (2,2-diphenyl-1-picrylhydrazyl) was prepared in methanol and homogenized in an ultrasonic bath. A working solution of the same concentration was used for analysis. From this solution, 1950 μL was mixed with 50 μL of the ethanolic extract. The mixture was vortexed at 2000 rpm for 30 s and incubated at room temperature in the dark for 30 min. The absorbance was measured at 517 nm.

Results from both assays were expressed as milligrams of Trolox equivalents per 100 g of dry weight (mg TE 100 g^−^^1^ DW).

### 4.9. Statistical Analyses

The experiment consisted of four replicates per treatment, with four plants allocated to each experimental unit. Chemical analyses were carried out in triplicate, and results are reported as mean value ± standard deviation. A one-way ANOVA was then conducted, with mean separation carried out by Duncan’s multiple-range test and significance defined as *p* < 0.05 (Tukey’s criterion). PCA and all other statistical procedures were performed in Statgraphics^®^ Centurion XVI (StatPoint Technologies, Warrenton, VA, USA).

## 5. Conclusions

This study demonstrates that electrical conductivity of the nutrient solution significantly influences the accumulation of health-promoting compounds in *S. tenerrimus*. A moderate EC level of 2.4 dS m^−1^ maximized the contents of phenolics and flavonoids and enhanced antioxidant activity, as measured by both ABTS and DPPH assays. In contrast, the lowest (1.2 dS m^−1^) and highest (3.0 dS m^−1^) EC levels resulted in reduced phytochemical concentrations, highlighting the sensitivity of this species to salinity extremes. The findings suggest that controlled salinity can serve as a practical tool to modulate the nutritional profile of *S. tenerrimus* grown in soilless systems. In relation to the spectrum of the lamps used, horticulture-specific lamps (L1, L2, and L3) exerted a pronounced effect on the growth and fertigation parameters of *S. tenerrimus* compared to the control (L4), whereas their influence on crop metabolites was limited or not clearly evident. This has potential applications for optimizing the functional value of wild edible greens in modern agriculture. For broader applicability, future research should investigate cultivar-dependent responses; long-term physiological effects; and interactions with other environmental variables such as light intensity, nutrient composition, and harvesting stage. These insights would support more targeted and sustainable cultivation strategies aimed at enhancing both crop resilience and human health benefits.

## Figures and Tables

**Figure 1 plants-14-02811-f001:**
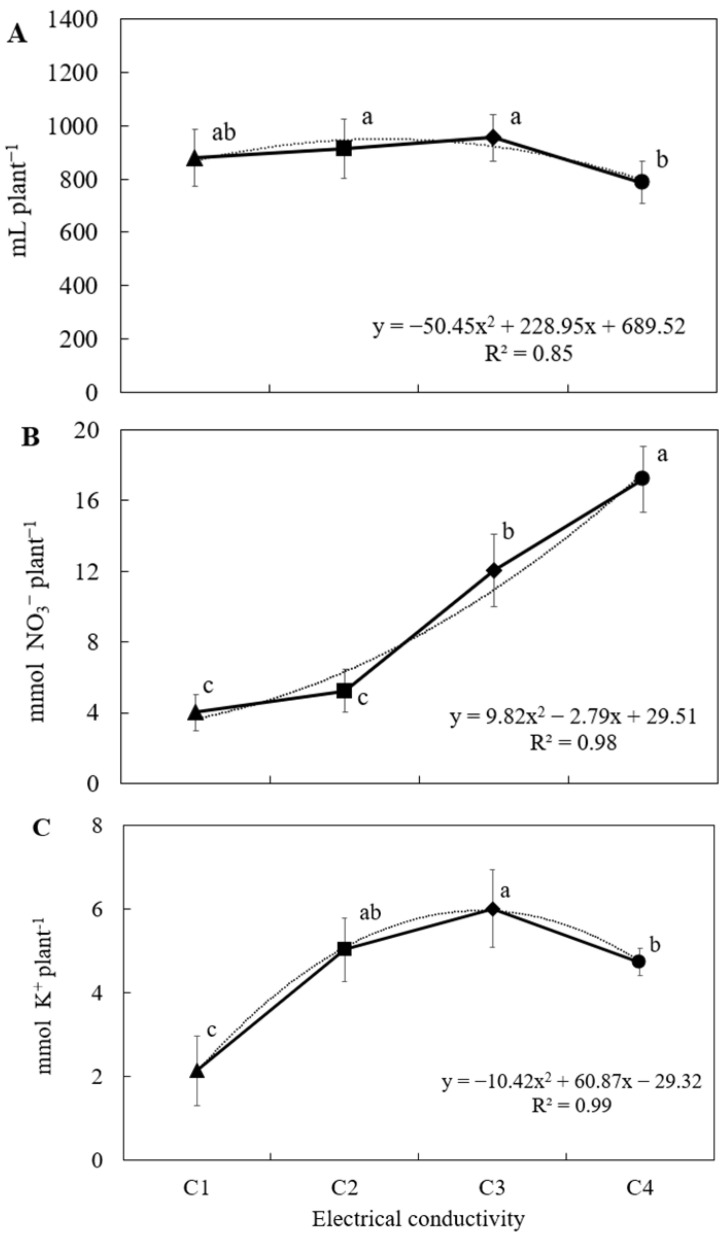
Water (**A**), nitrate (**B**), and potassium (**C**) uptake in relation to the electrical conductivity of nutrient solutions in *Sonchus tenerrimus* soilless culture at four salinity levels: C1, C2, C3, and C4, corresponding to 1.2, 1.8, 2.4, and 3.0 dS m^−1^, respectively. Different letters indicate statistically significant differences for each parameter (*p* < 0.05) according to Tukey’s test.

**Figure 2 plants-14-02811-f002:**
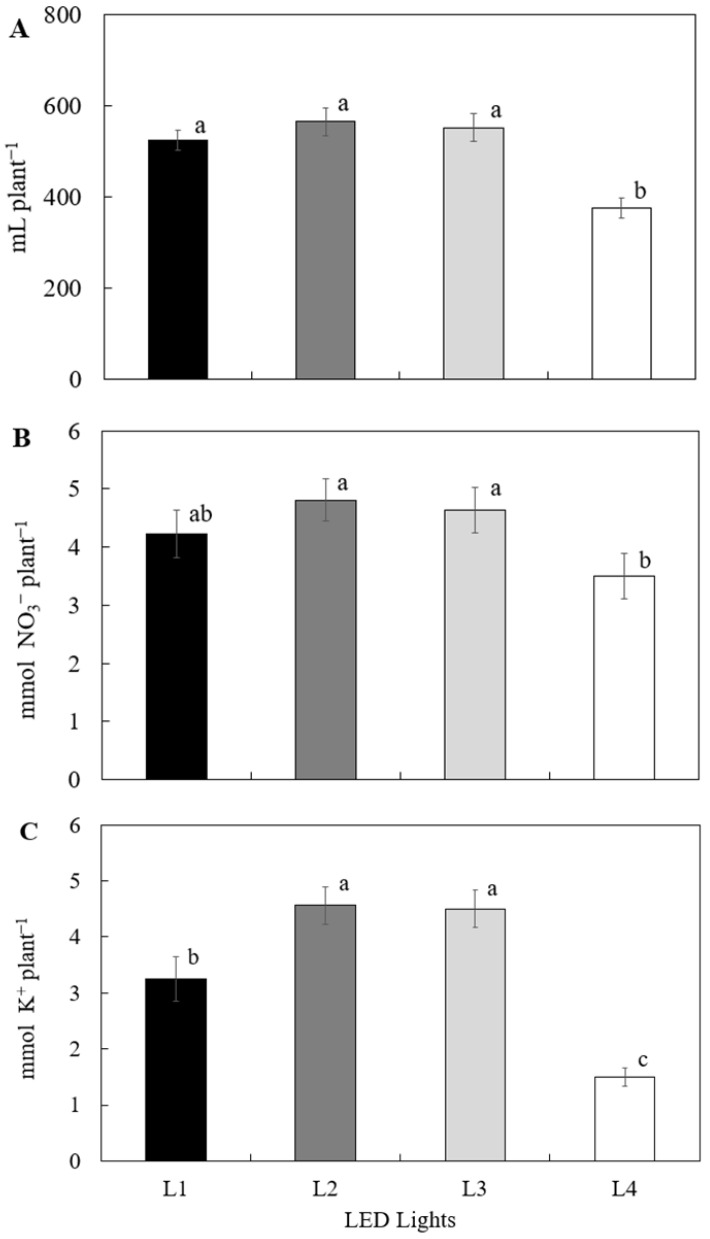
Water (**A**), nitrate (**B**), and potassium (**C**) uptake in relation to the light spectrum of the illumination system used in *Sonchus tenerrimus* soilless culture. Treatments: L1—L18 NS1 Valoya^®^, Helsinki, Finland; L2—L18 AP67 Valoya^®^; L3—L18 NS12 Valoya^®^; L4—L18 T8 Roblan^®^, Casarrubios del Monte Toledo, Spain. Different letters indicate statistically significant differences for each parameter (*p* < 0.05) according to Tukey’s test.

**Figure 3 plants-14-02811-f003:**
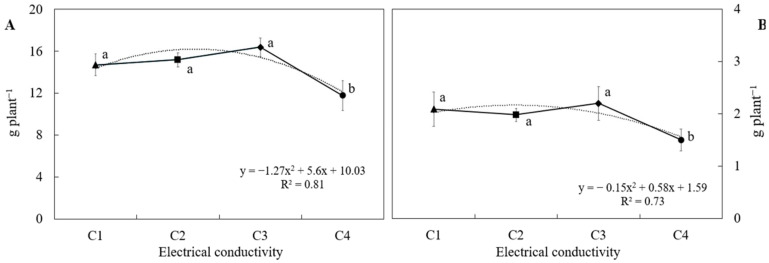
Fresh (**A**) and dry (**B**) biomass of *Sonchus tenerrimus* grown in soilless culture under different nutrient solution electrical conductivities: C1, C2, C3, and C4, corresponding to 1.2, 1.8, 2.4, and 3.0 dS m^−1^, respectively. Different letters indicate statistically significant differences for each parameter (*p* < 0.05) according to Tukey’s test.

**Figure 4 plants-14-02811-f004:**
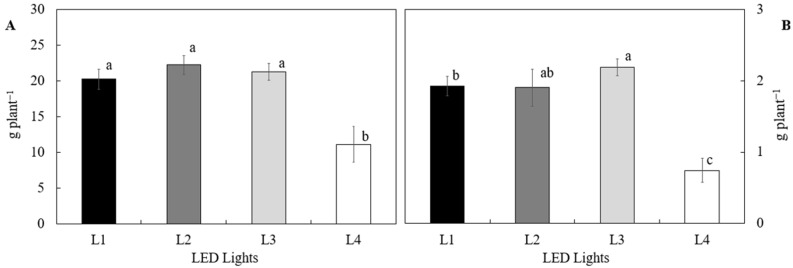
Fresh (**A**) and dry (**B**) biomass of *Sonchus tenerrimus* in soilless culture under different illumination spectra. Treatments: L1—L18 NS1 Valoya^®^; L2—L18 AP67 Valoya^®^; L3—L18 NS12 Valoya^®^; L4—L18 T8 Roblan^®^. Different letters indicate statistically significant differences for each parameter (*p* < 0.05) according to Tukey’s test.

**Figure 5 plants-14-02811-f005:**
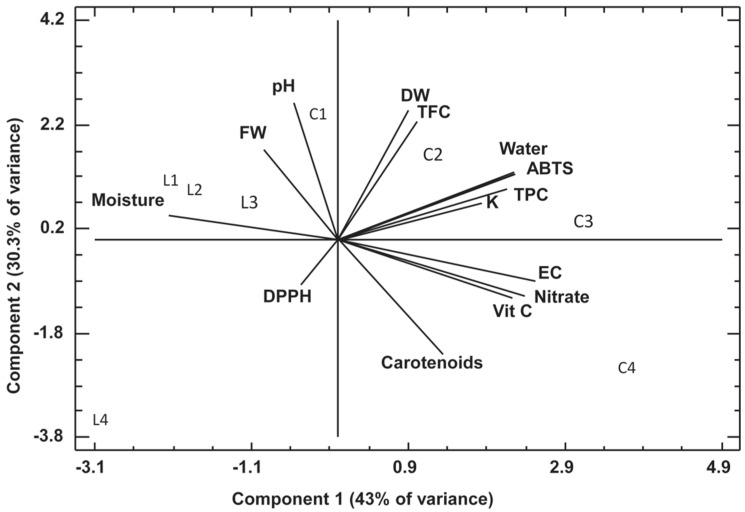
Biplot representing the first two principal components. Data points correspond to samples from the nutrient solution electrical conductivity (EC) trial (C1–C4) and the light-spectrum experiment (L1–L4), illustrating the distribution and relationships among treatments.

**Table 1 plants-14-02811-t001:** Data on moisture, total carotenoids, vitamin C, total phenolic, total flavonoids, and antioxidant activity of leaves from wild and cultivated *Sonchus tenerrimus* plants ^1,2,3,4^.

Samples/Codes	Moisture g 100 g^−1^	Total Carotenoids mg 100 g FW^−1^	Vitamin C mg 100 g^−1^ FW	TPC mg GAE 100 g FW ^−1^	TFC mg QE 100 g g FW ^−1^	DPPH mmol TE 100 g DW^−1^	ABTS mmol TE 100 g DW ^−1^
**Wild**							
WA	85.6 ± 0.6 ^bc^	61.5 ± 8.9 ^a^	69.3 ± 5.2 ^d^	358.8 ± 1.9 ^c^	119.5 ± 4.6 ^de^	1.8 ± 0.1 ^cde^	1.1 ± 0.0 ^de^
WD	87.6 ± 0.5 ^a^	45.3 ± 2.9 ^cd^	53.2 ± 8.8 ^f^	430.6 ± 10.2 ^a^	286.5 ± 9.6 ^a^	2.7 ± 0.1 ^ab^	0.9 ± 0.0 ^def^
WT	86.7 ± 0.5 ^abc^	59.6 ± 9.9 ^ab^	89.4 ± 2.1 ^bc^	250.9 ± 6.4 ^f^	180.1 ± 8.3 ^b^	2.0 ± 0.1 ^cd^	1.3 ± 0.1 ^de^
WU	87.0 ± 0.6 ^ab^	49.3 ± 3.7 ^bc^	54.0 ± 8.6 ^f^	360.7 ± 5.1 ^c^	139.7 ± 9.5 ^c^	2.1 ± 0.1 ^bcd^	0.8 ± 0.1 ^ef^
**Cultivated**							
Saline treatments							
C1 (1.2 dS m^−1^)	85.2 ± 0.3 ^c^	35.9 ± 1.0 ^de^	66.8 ± 4.3 ^de^	361.3 ± 6.5 ^c^	169.7 ± 6.6 ^b^	1.9 ± 0.3 ^cd^	2.5 ± 0.2 ^bc^
C2 (1.8 dS m^−1^)	85.3 ± 0.9 ^c^	38.7 ± 2.5 ^cde^	73.4 ± 1.4 ^d^	370.3 ± 15.2 ^bc^	166.2 ± 6.3 ^b^	1.5 ± 0.2 ^de^	2.9 ± 0.0 ^a^
C3 (2.4 dS m^−1^)	82.4 ± 0.5 ^d^	40.1 ± 4.4 ^cde^	97.0 ± 1.2 ^ab^	372.5 ± 14.4 ^bc^	122.1 ± 9.1 ^de^	2.2 ± 0.4 ^bc^	3.0 ± 0.1 ^a^
C4 (3.0 dS m^−1^)	82.5 ± 0.3 ^d^	48.2 ± 7.4 ^bc^	105.1 ± 3.6 ^a^	380.8 ± 6.0 ^b^	117.3 ± 3.5 ^de^	2.1 ± 0.7 ^bcd^	2.6 ± 0.1 ^ab^
Light treatments							
L1 (L18 NS1)	85.3 ± 0.8 ^c^	32.0 ± 6.6 ^e^	57.8 ± 1.3 ^ef^	284.7 ± 2.5 ^e^	126.1 ± 6.8 ^cd^	3.0 ± 0.4 ^a^	2.1 ± 0.5 ^c^
L2 (L18 AP67)	86.6 ± 0.0 ^abc^	34.9 ± 3.3 ^de^	67.5 ± 1.4 ^de^	305.6 ± 12.9 ^d^	108.9 ± 6.3 ^e^	1.9 ± 0.0 ^cd^	1.1 ± 0.3 ^de^
L3 (L18 NS12)	83.3 ± 1.7 ^d^	31.6 ± 0.7 ^e^	85.5 ± 5.9 ^c^	200.6 ± 3.4 ^g^	109.2 ± 11.6 ^e^	1.2 ± 0.3 ^e^	1.6 ± 0.2 ^d^
L4 (L18 T8)	86.0 ± 0.8 ^abc^	44.7 ± 1.5 ^cd^	70.5 ± 0.8 ^d^	197.0 ± 7.5 ^g^	72.1 ± 5.1 ^f^	2.4 ± 0.0 ^abc^	0.6 ± 0.0 ^f^

^1^ Data represent means ± standard deviation of samples analyzed in triplicate. ^2^ Differences in moisture, total carotenoids, vitamin C, total phenolics, total flavonoids, and antioxidant activity of the various samples were tested according to one-way ANOVA, followed by Duncan’s Multiple Range Test. ^3^ Within a column, means followed by different letters are significantly different at *p* < 0.05. ^4^ Results expressed on fresh weight (FW) as ascorbic acid (AA), gallic acid equivalents (GAEs), quercetin equivalents (QEs), and expressed on dry weight (DW) as Trolox equivalents (TEs).

**Table 2 plants-14-02811-t002:** Data on origin of seeds and plant status, collection location, and cultivation conditions of the *Sonchus tenerrimus* specimens used in this work.

Sample Code	Status	Collection Site of Seeds and Cultivation Conditions	EC ^1^ (dS m^−1^)	Lamp Type/Natural Light
WA	Wild	Almería, Almería (36.833252, −2.458631)	4.5	Natural light
WD	Wild	Aguadulce, Almería (36.809383, −2.574194)	3.0	Natural light
WT	Wild	El Toyo, Almería (36.836655, −2.318960)	4.2	Natural light
WU	Wild	University Campus, Almería, (36.828918, −2.405527)	2.2	Natural light
C1	Cultivated	University Campus of Almería, Growth chamber	1.2	L18 T8 Roblan^®^
C2	Cultivated	University Campus of Almería, Growth chamber	1.8	L18 T8 Roblan^®^
C3	Cultivated	University Campus of Almería, Growth chamber	2.4	L18 T8 Roblan^®^
C4	Cultivated	University Campus of Almería, Growth chamber	3.0	L18 T8 Roblan^®^
L1	Cultivated	University Campus of Almería, Growth chamber	2.0	L18 NS1 Valoya^®^
L2	Cultivated	University Campus of Almería, Growth chamber	2.0	L18 AP67 Valoya^®^
L3	Cultivated	University Campus of Almería, Growth chamber	2.0	L18 NS12 Valoya^®^
L4	Cultivated	University Campus of Almería, Growth chamber	2.0	L18 T8 Roblan^®^

^1^ EC: conductivity of the nutrient solution/soil of collection.

## Data Availability

The raw data supporting the conclusions of this article will be made available by the authors upon request.

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
