# Peer review of "Effect of Electrical Conductivity of Nutrient Solution and Light Spectra on the Main Phytochemical Content of *Sonchus tenerrimus* L. Under Wild and Controlled Environments"

_plants, 2025, doi:10.3390/plants14172811_

Round 1
Reviewer 1 Report
Comments and Suggestions for Authors
The research asseses the effects of two major abiotic factors, electrical conductivity (EC) and different LED light spectra on the growth parameters and phytochemical composition of a wild edible plant, which is of increasing interest for the increase in the productivity of such plants.
The authors have examined a wide range of ECs and light conditions with valuable results on yield and nutritional quality.
However there are some significant issues with the content. Specifically there are major inconsistencies with the indicated ECs (you mention 2.4 in abstract but this value is not mentioned anywhere in the text. Please check that the appropriate correct values are mentioned.
Also there is not clarity on which plants received the natural light treatment, so more details should be added to this
Something also very critical is the justification why you selected lamp L4 for the EC experiment as it was the worst of the 4, based on the LED light experiment. It needs to be explained.
You should also provide a more detailed discussion on the specific effects of the different LED lighting treatments on plant bioactive compounds.
Author Response
Reviewer
"... However there are some significant issues with the content. Specifically there are major inconsistencies with the indicated ECs (you mention 2.4 in abstract but this value is not mentioned anywhere in the text. Please check that the appropriate correct values are mentioned.
Authors:
All data related to the EC of the nutrient solution were reviewed and clarified in the new version.
"... Also there is not clarity on which plants received the natural light treatment, so more details should be added to this"
Authors:
All plants except those collected in the wild received artificial light, as clarified in Table 2.
"... Something also very critical is the justification why you selected lamp L4 for the EC experiment as it was the worst of the 4, based on the LED light experiment. It needs to be explained."
Authors:
We agree, but this is the spectrum more use by commercial grower
You should also provide a more detailed discussion on the specific effects of the different LED lighting treatments on plant bioactive compounds.
Authors:
This was considered and the discussion on this point was expanded (L. 399).
Reviewer 2 Report
Comments and Suggestions for Authors
Authors present their work focused how nutrient medium composition can affect growth and other parameters of slender sow thistle (SST) as well as effect of light. The outcomes of their research is overall interesting, but the missing Supplementary Materials, where authors have their results as well, decrease quality of the manuscript. Unfortunately, there are also other content and formal issues that should be edited or explained better.
Introduction
Line 36: names of plant families should be in same style (e. g. both in italic).
Line 56: ...chlorogenic, caffeic, luteolin, and luteolin-7-O-glucoside acids... - the last two compounds are flavonoids, not acids
Line 56: ...pinnatisect, and caualine... - these two terms refer to morphological character of leaves, not metabolites.
Line 62: WEP - please, write the abbreviation in full for the first time of use (wild edible plants, I suppose).
Line 107: ...in vitro.. - should be in italic. Same ...in vivo... on l. 513.
Results
Line 120 and furthermore: Supplementary Figures 1a-d, 2a-d are not included in Supplementary Materials.
Line 176: I would not rather use "linear" here, the curves have different progress.
Line 178: ...slight decrease... - according the Figure 1C, this decrease is significant in comparison with C3.
Line 193, Figure 1: Turkey's test is not mentioned in part 4.8.
Line 220: "In this study, the threshold was identified at an EC level of 4.5 dS m−1". According the data and text below, I would said that the the threshold was 3.5 dS m−1 as the last concentration caused the decrease of measured values.
Line 269: why was not content of other pigments measured?
Table 1: fw and dw - for other abbreviations, capital letters are used. There are also FW and DW in Figure 5.
Line 258: "Whitin...".
Line 258: results of statistical analyses are missing. In case, that there were not significant differences, the p- and F- values should be stated somewhere. The same applies to the graphs above.
Figure 5: line for carotenoids is blue - is it intentional?
Discussion
Generally, authors could more discussed the others effects of higher concentrations of salts, e.g. osmotic effect etc. that can participate on plants affecting.
This part of manuscript could contain more references on used Figures and Tables.
Line 450: Suaeda maritima, Sarcocornia fruticosa - missing abbreviation of names of authors.
Line 522: should not be: "In Figure 5..."?
Materials and Methods
Line 558: If possible, image of cultivation system could be added in supplementary materials as well.
Line 559: Which specific information do authors mean? Table 1 shows rather results.
Line 568: here, the different conductivities are stated (3.0, 4.0, 6.0 dS. m-1) .
Table 2: how was EC measured in case of wild plants?
Line 581: Ice plant leaves - should not be SST?
Line 586: it was stated in previous section that plants was dried. When was the fresh material for carotenoids, vit. C, etc. collected? Were the analyses performed immediately or what were conditions of storage?
Line 606: 250 mm x 4.6 mm. There should be "×".
Line 616: which material was used - dry or fresh?
Line 620: chromatography is not described for subsequent analyses.
Line 625: Most of Supplemental Files are missing.
Line 652: ...an antioxidant...
References
There are 16 references, where articles of one or more authors of this manuscript are used. Thus, it is over 20 % of used references, when it should be under 15%.
Supplementary Materials
Name of article in available Supplementary material differs from main manuscript.
Growth, fertigation, and lighting conditions applied to cultivated Sonchus tenerrimus
Where did authors acquired seeds/plants for cultivated status?
Used EC and labeling of "L" variants are different in comparison with main text.
Why were not the results of from measurement of plant parts in the main text?
Plants were collected at baby leaf stage (16-17 leaves) - is it also true for wild plants?
Some of cited sources in text are not in the references - Urrestarazu et al., 2008.
Author Response
Reviewer
"... Line 36: names of plant families should be in same style (e. g. both in italic).
Authors:
Changed
"... Line 56: ...chlorogenic, caffeic, luteolin, and luteolin-7-O-glucoside acids... - the last two compounds are flavonoids, not acids
Line 56: ...pinnatisect, and caualine... - these two terms refer to morphological character of leaves, not metabolites.
Authors:
Corrected
"... Line 62: WEP - please, write the abbreviation in full for the first time of use (wild edible plants, I suppose).
Authors:
Added
".... Line 107: ...in vitro.. - should be in italic. Same ...in vivo... on l. 513.
Authors:
Corrected
"... Line 120 and furthermore: Supplementary Figures 1a-d, 2a-d are not included in Supplementary Materials.
Authors:
We have added the figures 1, 2 and 3, to the supplementary material. The authors have uploaded the complete supplementary files, but the publisher may not provide them to you. In any case, we have uploaded them here as well.
Line 176: I would not rather use "linear" here, the curves have different progress.
Authors:
We agree, this was corrected
Line 178: ...slight decrease... - according the Figure 1C, this decrease is significant in comparison with C3.
Authors:
This was corrected in the new version.
“… Line 193, Figure 1: Turkey's test is not mentioned in part 4.8.
Authors:
We have corrected in the new version.
“… Line 220: "In this study, the threshold was identified at an EC level of 4.5 dS m−1". According the data and text below, I would said that the the threshold was 3.5 dS m−1 as the last concentration caused the decrease of measured values.
Authors:
We have corrected in the new version.
“… Line 269: why was not content of other pigments measured?
Authors:
We focused our analysis on carotenoids because they are the main pigments related to light responses and nutritional quality in the studied species. Other pigments, such as anthocyanins or chlorophylls, were not quantified since they were outside the scope of our research question, which aimed to evaluate the impact of light treatments specifically on carotenoid accumulation. However, we agree that additional pigment analyses could provide a more complete picture of the plants’ physiological responses, and we will consider this in future studies.
“… Table 1: fw and dw - for other abbreviations, capital letters are used. There are also FW and DW in Figure 5.
Authors:
We have changed everything to capital letters
“… Line 258: "Whitin...".
Authors:
Corrected
“… Line 258: results of statistical analyses are missing. In case, that there were not significant differences, the p- and F- values should be stated somewhere. The same applies to the graphs above.
Authors:
This was corrected in the new version. This was a mistake
“… Figure 5: line for carotenoids is blue - is it intentional?
Authors:
This is a mistake when converting to TIFF, which has been corrected in the new version.
Discussion
“… Generally, authors could more discussed the others effects of higher concentrations of salts, e.g. osmotic effect etc. that can participate on plants affecting.
This part of manuscript could contain more references on used Figures and Tables.
Authors:
Somes comments were enclosed in relation to Salinity and like this affects to plant
“… Line 450: Suaeda maritima, Sarcocornia fruticosa - missing abbreviation of names of authors.
Authors:
We have incorporated the abbreviations in the revised version.
“… Line 522: Shouldn't it read: "In Figure 5..."?
Authors:
We have added in the new version.
Materials and Methods
“... Line 558: If possible, image of cultivation system could be added in supplementary materials as well.
Authors:
We partially agree, but we have decided not to expand the manuscript, which is already sufficiently long according to the editors.
“… Line 559: Which specific information do authors mean? Table 1 shows rather results.
Authors:
We have corrected the text in the new version, it is in the Table 2 and the information is collection location and cultivation conditions of Sonchus tenerrimus specimens used in this work.
“… Line 568: here, the different conductivities are stated (3.0, 4.0, 6.0 dS. m-1)
Authors:
We have corrected this information in the new version.
“…. Table 2: how was EC measured in case of wild plants?
Authors:
The EC of samples collected from plants in the wild is not irrigated with nutrient solution; only its EC in the saturated extract of natural soil is measured.
“… Line 581: Ice plant leaves - should not be SST?
Authors:
We have corrected this information in the new version. This was a mistake.
“…. Line 586: it was stated in previous section that plants was dried. When was the fresh material for carotenoids, vit. C, etc. collected? Were the analyses performed immediately or what were conditions of storage?
Authors:
We have added the information in the new version.
“…. Line 606: 250 mm x 4.6 mm. There should be "×".
Authors:
Corrected
“… Line 616: which material was used - dry or fresh?
Authors:
The analyses (as described in Materials and Methods) were performed on dry matter, although the results were extrapolated to fresh matter.
“…. Line 620: chromatography is not described for subsequent analyses.
Authors:
This was considered in the new version
“... Line 625: Most of Supplemental Files are missing.
Authors:
We have sent all supplementary files to the journal. See comment above
“…. Line 652: ...an antioxidant...
Authors:
Corrected
References
There are 16 references, where articles of one or more authors of this manuscript are used. Thus, it is over 20 % of used references, when it should be under 15%.
Authors:
We agree with the comments. We have reviewed all the references in the manuscript and have removed approximately half of them.
Supplementary Materials
“… Name of article in available Supplementary material differs from main manuscript. Growth, fertigation, and lighting conditions applied to cultivated Sonchus tenerrimus
Authors:
We have corrected in the new version.
“…. Where did authors acquired seeds/plants for cultivated status?
Authors:
This was clarified in the Table 2
“…. Used EC and labeling of "L" variants are different in comparison with main text.
Authors:
We have corrected all the information in the new version.
“… Why were not the results of from measurement of plant parts in the main text?
Authors:
This would be possible and interesting, but we have considered the edible plant because in other to simplify the manuscript
“…. Plants were collected at baby leaf stage (16-17 leaves) - is it also true for wild plants?
Authors:
We have corrected this information in the new version.
Some of cited sources in text are not in the references - Urrestarazu et al., 2008
Authors:
We have corrected in the new version.

Reviewer 3 Report
Comments and Suggestions for Authors
The manuscript describes the effect of electrical conductivity of nutrient solution and light spectra on the main phytochemical content of Sonchus tenerrimus L. under wild and controlled environments.
The language is fluent. Please, include all amendments in the revised manuscript.
- Ls.17-18: Unclear.
- L.62: Please, provide full name of WEP.
- Ls.263-4: Not true, see Table 1.
- L. 292: Please, provide full name for WAG.
- Ls.334-4: The pH change should be justified.
- Ls. 343-4: Explain why.
- Ls. 390-401: Trivial; the novelties of this study should be highlighted.
- Ls. 448-53: Unclear.
Author Response
“…. Ls.17-18: Unclear.
Authors
We have clarified in the new version.
“…. L.62: Please, provide full name of WEP.
Authors
We have clarified the new version.
“…. Ls.263-4: Not true, see Table 1.
Authors
We have changed the new version.
- 292: Please, provide full name for WAG.
Authors
We have clarified in the new version.
Ls.334-4: The pH change should be justified.
Authors
This was justified in the new version.
“... Ls. 343-4: Explain why.
Authors
Lighting affects plant physiology, but drainage pH and EC are primarily determined by nutrient solution chemistry and substrate interactions. If these factors remain consistent, changes in lighting alone are unlikely to cause significant differences in the chemical parameters of the drainage.
“…. Ls. 390-401: Trivial; the novelties of this study should be highlighted.
Authors
Some comments in conclusions were enclosed
“.... Ls. 448-53: Unclear.
Author:
This was rewritten

Round 2
Reviewer 1 Report
Comments and Suggestions for Authors
The amendments by the authors significantly improved the manuscript
Author Response
We agree with this status
Reviewer 2 Report
Comments and Suggestions for Authors
Authors edited the second version of their manuscript according to the most of comments and suggestions. However, there are still some parts that could be more explained.
Introduction
Line 41 - S. asper and S. arvensis - please add abbreviation of authors.
Results
Line 127 - This comment is related to missing picture of the growing system and additional information regarding cultivation in Material and Methods/Suppl. Mat. Besides growing conditions, please add information how the cultivating system looked (hydroponic basins, pots, the volume etc.). I suppose that authors had applied nutrients in solution only once on the beginning of experiment, but was evaporated water also refilled? If the salts/nutrients were not fully absorbed, the EC should be high on the start as well. On the other hand, in case of some absorption, I would expect its decrease on the beginning at least. Could authors provide more explanation about this effect here or in the discussion?
Line 260, 261 - (FW), (DW) instead of (fw), (dw)
Line 293 - WAG
Discussion
This part of manuscript could still contain more references on used Figures and Tables (e. g. Fig 1, Table 2 etc. in corresponding text).
Materials and Methods
Table 2 - I did not find origin of cultivated plants (e. g. provider of seeds) in this Table.
Author Response
Introduction
Line 41 - S. asper and S. arvensis - please add abbreviation of authors.
It was written Sonchus both in the new version
Results
Line 127 - This comment is related to missing picture of the growing system and additional information regarding cultivation in Material and Methods/Suppl. Mat. Besides growing conditions, please add information how the cultivating system looked (hydroponic basins, pots, the volume etc.). I suppose that authors had applied nutrients in solution only once on the beginning of experiment, but was evaporated water also refilled? If the salts/nutrients were not fully absorbed, the EC should be high on the start as well. On the other hand, in case of some absorption, I would expect its decrease on the beginning at least. Could authors provide more explanation about this effect here or in the discussion?
Part of this comments were enclosed in suplementaries files. As the article was considered too lengthy, part of the content was placed in the supplementary material. As:
«The application of a new fertigation” In supplementary files”
And “…The plants were grown in one-liter pots in a soilless cultivation system using coconut fiber. The physical and physicochemical characteristics of the coconut fiber were described by Peçanha et al. [19].” In supplementary files”
In traditional fertigation of soilless crops, the electrical conductivity of the nutrient solution is always higher under growing conditions. Therefore, in the revised text it is justified that this value is one unit higher than the input, which is compensated by the new fertigation strategy (as explained in the new supplementary file) with 20% drainage.
Line 260, 261 - (FW), (DW) instead of (fw), (dw)
Done and all new version
Line 293 – WAG
Done
Discussion
This part of manuscript could still contain more references on used Figures and Tables (e. g. Fig 1, Table 2 etc. in corresponding text).
Several figure and table citations were incorporated in the revised version.
Materials and Methods
Table 2 - I did not find origin of cultivated plants (e. g. provider of seeds) in this Table.
This was clarified in the new version (Table 2).

Round 3
Reviewer 2 Report
Comments and Suggestions for Authors
The text was edited accordingly, but by adding abbreviation of authors, it was rather meant that there should be Sonchus asper (L.) Hill and Sonchus arvensis L. Thank you for clarification regarding origin of seeds, it would be sufficient to state they were collected from wild plants growing at the campus in that case.
Author Response
The text was edited accordingly, but by adding abbreviation of authors, it was rather meant that there should be Sonchus asper (L.) Hill and Sonchus arvensis L. Thank you for clarification regarding origin of seeds, it would be sufficient to state they were collected from wild plants growing at the campus in that case
Author:
Thank you for your comments ".. Sonchus asper (L.) Hill and Sonchus arvensis L." was rewritten